



# Analysis of the effects of urban micro-scale vulnerabilities on tsunami evacuation using and Agent-Based model. Case study in the city of Iquique, Chile

Rodrigo Cienfuegos[1,2], Gonzalo Álvarez[1], Jorge León[3], Alejandro Urrutia[2], and Sebastián Castro[2]

[1]Departamento de Ingeniería Hidráulica y Ambiental, Escuela de Ingeniería, Pontificia Universidad Católica de Chile
[2]Centro de Investigación para la Gestión Integrada del Riesgo de Desastres - CIGIDEN -ANID/Fondap/1522A0005
[3]Departamento de Arquitectura, Universidad Técnica Federico Santa María

**Correspondence:** Rodrigo Cienfuegos (rcienfue@uc.cl)

**Abstract.** The occurrence of mega tsunamis over the last couple of decades has greatly increased the attention of the research community and practitioners to work hand in hand to reduce risks from these highly destructive threats. Protecting the population through evacuation is the best alternative for avoiding loss of life in the wake of the occurrence of a tsunamigenic earthquake. Therefore, guaranteeing the proper state of evacuation routes is very important to ensure appropriate movement to the safe zones. This study carries out a detailed analysis of possible evacuation scenarios, considering the actual state of the escape routes of Iquique, a Chilean city prone to tsunamis, with a substantial amount of urban micro-scale vulnerabilities, i.e., elements that obstruct or complicate pedestrian flow. The quantification of the delay in evacuation processes resulting from the presence of urban micro-vulnerabilities is carried out through micro-scale Agent-Based Modelling (ABM). In addition, these results are integrated with high-resolution tsunami inundation simulations, allowing for an estimation of the potential number of people that the tsunami may reach under different scenarios, by emulating the dynamics and behavior of the population and the decision making regarding the starting time of the evacuation.

## 1 Introduction

During recent decades, giant tsunamis have left thousands of casualties in coastal areas. The deadliest among them was the tsunami that occurred after the Mw 9.0 Indian Ocean earthquake in 2004. It is estimated that over 200,000 people died as a result of the tsunami, which reached the coasts of Sumatra and Sri Lanka approximately 30 minutes after the earthquake (Puspito and Gunawan, 2005; Rabinovich et al., 2011). Tsunami research has come a long way after this event and numerous advancements have been achieved (e.g. Satake, 2014a; Okal, 2015). It is now possible to estimate earthquake parameters and their tsunamigenic potential using faster and more robust methodologies (e.g. Gusman and Tanioka, 2014; Melgar and Bock, 2015; Crowell et al., 2016). Tsunameter networks for direct sea surface observations, both in deep water and on the coast, have also been expanded (e.g. Wächter et al., 2012; Mulia and Satake, 2020; Catalan et al., 2020). In addition, new tsunami forecasting models, constrained with nearly real-time data assimilation methods are now available to provide a fast and accurate assessment of tsunami waves propagation (e.g. Maeda et al., 2015; Wang et al., 2017; Navarrete et al., 2020). Early



warning systems have been accompanied by other measures in tsunami-prone areas, including hazard mapping, education and awareness programs, the installation of speakers to broadcast alerts, signage indicating evacuation routes, and in some cases,
the construction of hard infrastructures such as large breakwaters (Shuto and Fujima, 2009; Satake, 2014b).

After the Mw 9.0 Great East Japan Tsunami in 2011, wave heights were underestimated in the first broadcasted alarm bulletin issued 3 minutes after the nucleation of the earthquake (Cyranoski, 2011). The tsunami protection hard infrastructure was overwhelmed, with much of it destroyed, and unable to stop the tsunami with deadly consequences (Suppasri et al., 2013b). The failure of the structural defenses was a reminder that they are insufficient to protect the exposed population, and
must be combined with soft countermeasures such as education and awareness for promoting a quick evacuation to safe zones (Koshimura and Shuto, 2015).

The 2010 Mw 8.8 Maule earthquake and tsunami in Chile resulted in 525 deaths, 156 of which were attributed to the tsunami (Huerta, 2011). In this event, the tsunami alarm was not broadcasted promptly by local authorities, but the death toll was minimized by the quick self-evacuation carried out by the residents of coastal communities thanks to their knowledge of
past events (Fritz et al., 2011). The 2010 earthquake was the first of 3 tsunamigenic events that occurred along the coast of Chile in less than 6 years; it was followed by the Mw 8.2 earthquake of April 1st 2014 in Iquique (Catalán et al., 2015; Tomita et al., 2016), and the Mw 8.3 earthquake of September 16th 2015 in Illapel (Aránguiz et al., 2016; Contreras-López et al., 2016). Despite their large magnitudes, these events only caused a few casualties thanks to the good performance of building codes, and the rapid evacuation of the population after the shake; however damage to port infrastructure, households in coastal
settlements, and fisherman's coves were produced by the ensuing tsunamis.

Avoiding loss of life is the main objective of risk mitigation, and tsunami evacuation has proved to be the most effective countermeasure to keep the population safe (Shuto, 2005; Suppasri et al., 2013a), especially in countries where hard counter-measures have not been widely adopted due to their high costs. Investment in risk awareness, education, and urban planning and infrastructure improvements are vital initiatives for ensuring that evacuation processes are carried out successfully (Scheer
et al., 2012; Esteban et al., 2013). Among these, the urban built environment is an important dimension to consider for evacuation planning, since it provides spaces for evacuation and mobility. An appropriate urban design contributes to risk reduction and enhances proper reactions and attitudes of people during events such as earthquakes (Ciborowski, 1982). Evacuation routes, which allow population displacement in search of refuge, are particularly susceptible to increases in vulnerability. The works of León and March (2016) and Reyes and Miura (2016) suggest the need to carry out a detailed analysis of these public spaces
that are used during evacuation processes. Addressing this problem, Álvarez et al. (2018) present a method to identify urban micro-vulnerabilities along evacuation routes, showing that the presence of these elements at the pedestrian-experience scale can negatively affect the displacement of evacuees during their escape.

Studying the behavior of the population during an evacuation is difficult as the observation and measurement of these processes are generally possible only during drills; however, halting the normal activities of a city to carry them out is not
something that can be done regularly. Although the importance of drills in preparedness and awareness of the population is undeniable, they do not allow modifications to analyze a greater number of variables, as needed in scientific experimentation. To this end, evacuation models have been broadly developed during recent decades and have been used in a series of applications





such as crowd dynamics, pedestrian movement, and human behavior during emergencies (e.g. Bellomo et al., 2016). Evacuation modeling seeks to get as close as possible to reproducing these processes and allows the decisions of designers and legislators to be guided (León et al., 2021). Among the multiple functions of evacuation modeling, the possibility of assessing the interaction of evacuees with the built environment opens the possibility of analyzing the urban form and its potential improvements. This work seeks to contribute in this regard; through micro-scale evacuation modeling integrated with high-resolution tsunami flooding simulations, it examines how the evacuation route environment influences these processes. The study was carried out in the coastal city of Iquique, which was affected by a tsunami in 2014.

The article is organized as follows: we first summarize the evidence of high seismic hazard in the study area suggesting the possible existence of a high risk of tsunami evacuation; secondly, we present the models and computational tools that we use to study tsunami evacuation processes; then the main results and a discussion of their implications are presented, followed by concluding remarks and recommendations.

## 2   Study area

The Peru-Chile Trench is known for being the zone of convergence between the Nazca and South American plates having a high seismic activity. The largest reported earthquakes in recent centuries in the area are the 1868 Arica and 1877 Iquique earthquakes, both of magnitudes estimated around Mw 8.8 and followed by destructive tsunamis (Lomnitz, 2004). The so-called "Norte Grande" of Chile, located between the Chile-Peru border and the peninsula of Mejillones near the city of Antofagasta (Figure 1), has received special attention owing to the existence of a seismic gap with high accumulation of elastic deformation (Comte and Pardo, 1991; Chlieh et al., 2011; Métois et al., 2013). Even if destructive earthquakes have been nucleated in the area over the last 60 years (Mw 7.4 1967 and Mw 7.7 2007 near Tocopilla, and the Mw 8.2 Iquique earthquake in 2014), they represent only a small fraction of the accumulated deformation (Catalán et al., 2015; Yáñez-Cuadra et al., 2022) which means that there is enough potential seismic energy to produce a major tsunamigenic subduction earthquake (Hayes et al., 2014; González et al., 2020).

The seismic context described above motivates us to place our study in the city of Iquique, a port city located in the north of Chile (20.53° S, 70.18° W). We focus on the downtown zone which consists of a narrow strip no wider than 3 km from east to west with an upward slope that ends in the coastal mountain range. Iquique hosts intense industrial and commercial activity due to the presence of one of the most important ports in the country, which is used for the transport of goods and mining resources. Among its attractions are one of the largest duty-free zones in South America, historical buildings, the natural beauty of its beaches and a year-round temperate climate. The city has undergone major growth, with a rate of around 70% between 1993 and 2003. According to the results of the last census, a total of 184,953 people reside in the city (INE, 2012). The rapid growth of coastal cities such as Iquique has led to increased exposure of their inhabitants and property to hazards such as tsunamis and a consequent increase in vulnerability and flood risk (Neumann et al., 2015; Kron, 2013; Jongman et al., 2012).


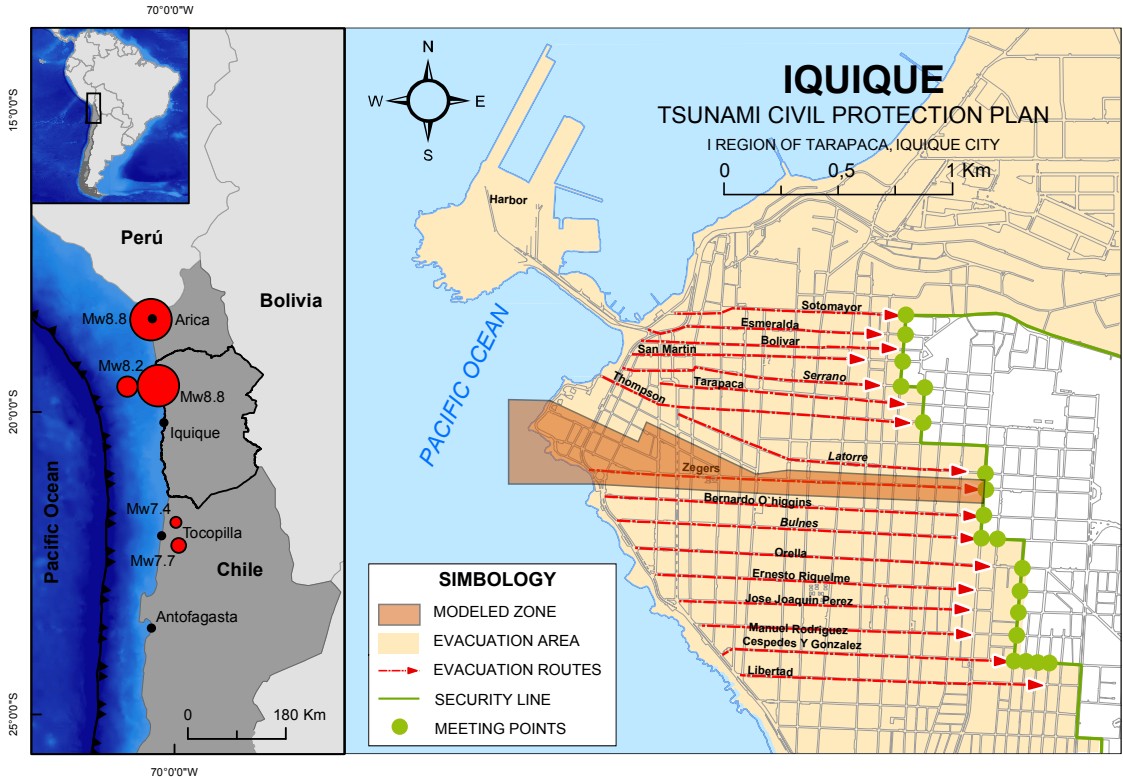

**Figure 1.** Left panel: Geographical location of the study area and major earthquakes (red circles) reported in Chile since 1868 (epicenters based on the records of the National Seismological Center of the University of Chile and the United States Geological Survey). Right panel: Downtown Iquique and the tsunami evacuation routes where the Zegers street is highlighted (the map has been modified from ONEMI and IMI (2013)).

## 3 Research methods and modeling tools

### 3.1 Urban micro-vulnerabilities along evacuation routes

During an evacuation, escape routes to higher ground are determining factors for the proper execution of the process. An appropriate evacuation route layout ensures that the path of travel to safe areas is the shortest; in Iquique the orthogonal street layout is a positive element that guarantees redundant and practically straight evacuation routes (see Figure 1). Nonetheless, the high population growth, vehicle ownership rates, and large commercial, tourism, industrial and educational activities in the study area, increase the tsunami risk. High occupancy rates give rise to a series of problems in terms of space availability along evacuation routes, which ideally should remain clear and free of obstacles to avoid a decrease in their design capacity (Scheer et al., 2012).


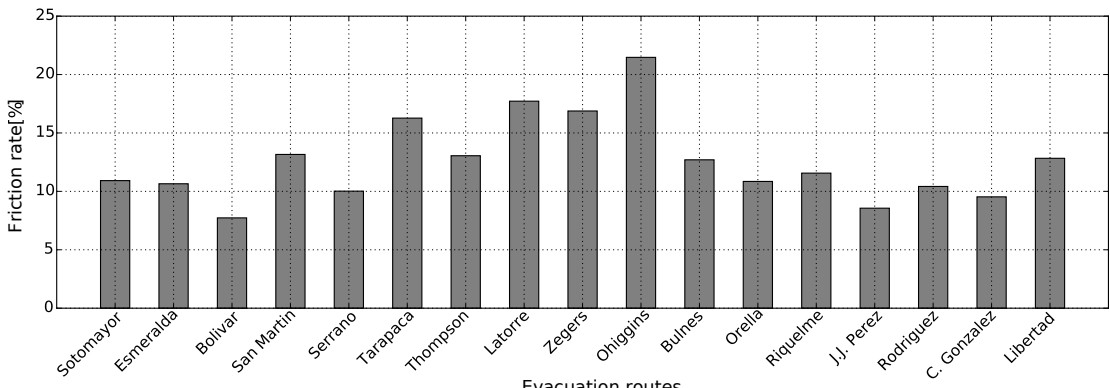

**Figure 2.** Friction rates of the main evacuation routes of Iquique downtown, based on the results of Álvarez et al. (2018).

Iquique downtown shows many of these complications that can negatively affect the displacement of evacuees at the pedestrian-experience scale (León and March, 2016; Walker, 2013; Álvarez et al., 2018). More specifically, the work of Álvarez et al. (2018) documents the presence of 'urban micro-scale vulnerabilities' along evacuation routes that stem from the poor use, maintenance or design problems, and propose a methodology to quantify their effect on the displacement of evacuees through an urban friction rate. Another important issue is the use of public spaces for parking, which considerably reduces the route capacity; the latter is due to the great number of vehicles in circulation in Iquique, with 3.62 vehicles per 10 inhabitants, much higher than the national average of 2.64 (INE, 2015). While other factors such as the use of sidewalks to extend the service areas of restaurants and the use of public spaces for informal commerce have a smaller presence, they similarly slow evacuations. The analysis of tsunami evacuation scenarios developed in Álvarez et al. (2018) allows to quantify the presence of urban micro-vulnerabilities along the evacuation routes of Iquique to realistically represent evacuation processes in light of the city's current conditions. Figure 2 shows the results of the assessment of the evacuation routes of Iquique (their locations are depicted in Figure 1) as a function of their friction rates; the routes with urban environment that are the worst prepared for an eventual evacuation are O'Higgins, Latorre, and Zegers streets. Among them, in what follows we focus on Zegers Street to carry out a detailed analysis of evacuation owing to the rapid tsunami arrival and flooding along it.

### 3.2 Tsunami modeling

In this study, numerical simulations of hypothetical tsunami scenarios are developed using the open-source GeoClaw model (Berger et al., 2011). The GeoClaw code solves the nonlinear shallow water equations using a mesh-adaptive high-resolution finite volume method to compute spatio-temporal evolution of water depths $h(x, y, t)$ and depth-averaged flow velocities in horizontal dimensions, $u(x, y, t)$ and $v(x, y, t)$. It has been widely applied for tsunami hazard assessment in different topo-bathymetric contexts (e.g. Melgar and Bock, 2015; Cienfuegos et al., 2018; Williamson et al., 2020).

The tsunami scenarios are modeled using four nested grids in planar coordinates to achieve a high-resolution description of the tsunami propagation and inundation from its generation zone into the city (see Figure 3). Grid 1 corresponds to the
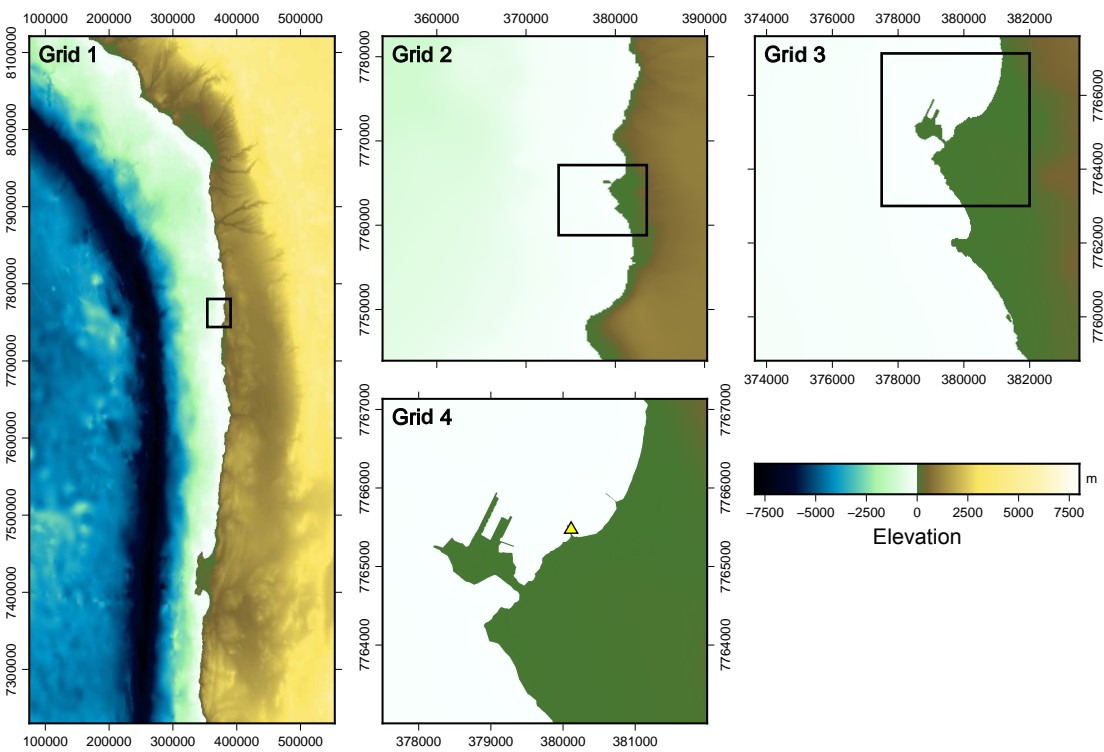

**Figure 3.** Nested grids used in the numerical simulations. The yellow triangle indicates the location of the Port of Iquique tide gauge.

bathymetry of the Pacific Ocean, generated from the GEBCO 2014 digital global bathymetric model (Weatherall et al., 2015), which covers the tsunami generation zone with a coarse resolution of 900 m. Grids 2 and 3, with resolutions of 180 and 30 m, respectively, are obtained from the nautical charts of the National Hydrographic and Oceanographic Service of the Chilean Navy (SHOA), to propagate the tsunami to the Chilean coast. The inundation modeling is performed with Grid 4 using LIDAR topographic information of the city of Iquique with a resolution of 2 m (provided by the JICA-JST Tsunami project, see Tomita
et al. (2016)), to achieve a high level of detail at the street level.

   The validation of the model configuration is performed using tsunami observations of the 2014 Iquique earthquake. We compare computed tsunami signals with tidal gauge records at the Iquique port and with DART buoys (UNESCO/IOC, 2014; Heidarzadeh et al., 2015) using two source models. The first one is obtained from the inversion of tsunami records (An et al., 2014), while the second one incorporates in addition GPS measurements to perform the source inversion (Gusman et al., 2015).
The initial conditions for the tsunami simulations are estimated based on the displacement distribution of the earthquake source using the elastic dislocation theory of Okada (1985).

   In general the deep water DART tsunami time series generated from the fault models of An et al. (2014) and Gusman et al. (2015) are in good agreement with the tsunami records (Figure 4). Nevertheless, an earlier tsunami wave arrival is


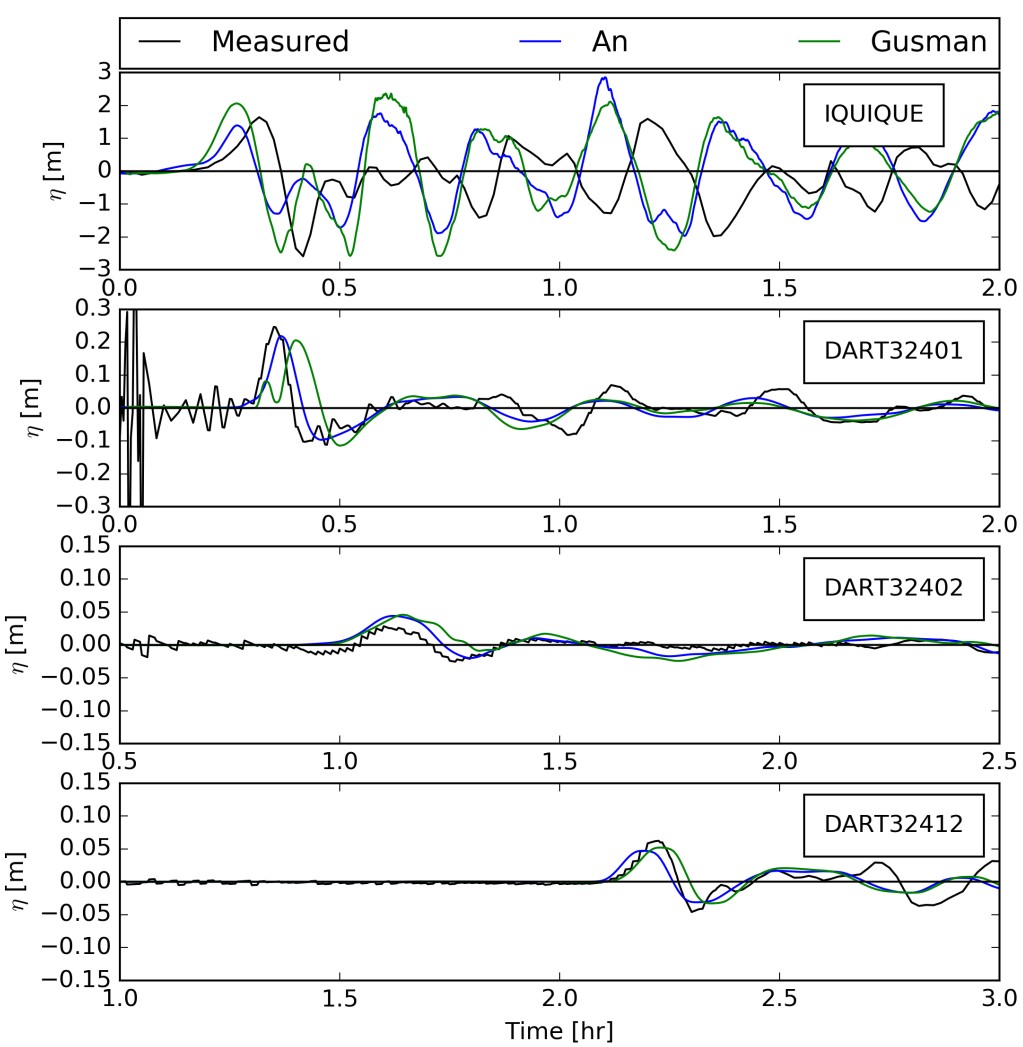

**Figure 4.** 2014 tsunami signals recorded by the Iquique tide gauge and DART buoys (black), compared with time series generated by the numerical simulation using An et al. (2014) (blue) and Gusman et al. (2015) (green) earthquake sources.





observed in the simulations at the Iquique tide gauge, as was also the case in simulations reported in Catalán et al. (2015) and

Gusman et al. (2015). Some differences in phases and amplitudes are also observed at this tidal gauge, consistent with results reported previously by the same authors. These differences might be due to the inherent epistemic uncertainty associated with the modeling of rupture inversion and its consequent variability in finite fault models (Cienfuegos et al., 2018), and/or an insufficient bathymetric resolution and information to properly represent the port layout, which may result in a poor description of the later stages of the tsunami evolution. Nonetheless, model results are in agreement with previously published works

dealing with the same event, and are thus considered adequate to be applied in the present study.

### 3.3 Tsunami scenarios

We adopt the Interseismic Coupling model (ISC) approach (Li et al., 2015) to define potential future earthquake slip sources in the Norte Grande seismic gap. For the tsunami evacuation analysis, we define two cases: a Mw 9.0 earthquake based on the ISC estimations of Chlieh et al. (2011), and a Mw 8.4 earthquake from the large sample of stochastic scenarios developed by

Métois et al. (2013) selected for its high flood level and early arrival time. Both scenarios were proposed as possible earthquakes before the 2014 Iquique earthquake, thus giving a conservative assessment of potential ruptures. The tsunami simulations are performed assuming a mean tide level and a high tide level (+ 0.76 m) at the time of the nucleation of the earthquake.

### 3.4 Evacuation modeling

The evacuation model used herein is based on the one validated by Poulos et al. (2018) which consists in an Agent-Based Model

(ABM) to statistically aggregate the evacuation displacements from the individual level. The model uses a collision avoidance algorithm (Van Den Berg et al., 2011) which naturally captures congestion problems that may occur during an evacuation. We define a 45-cm agent space where no agent overlapping is allowed. The Dijkstra shortest path algorithm (Dijkstra, 2022) is set as the evacuation route selection. The displacement velocity of each agent is sampled from a Weibull distribution with a mean of 1.34 m/s (shape parameter 10.14 and scale parameter 1.41), calibrated by Rinne et al. (2010) and consistent with the

literature review conducted by Daamen and Hoogendoorn (2007). The change in pedestrian velocity as a function of the slope of the terrain is calculated using Tobler's hiking function (Tobler, 1993), which has been employed in similar studies (Wood and Schmidtlein, 2012; Fraser et al., 2014; Solís and Gazmuri, 2017).

A realistic representation of the urban environment is obtained by including the urban micro-vulnerabilities present along the studied evacuation route following the methodology and field data of Álvarez et al. (2018). The effect that these features

have on the agent movement is taken into account using the Speed Conservation Value (SCV) (Schmidtlein and Wood, 2015; Fujiyama and Tyler, 2004), which is the fraction of the maximum speed that a pedestrian can reach on a given surface (Wood and Schmidtlein, 2012). The SCV associated to urban micro-vulnerabilities are either blockages where the agents cannot go through (SCV=0), partial speed reductions when they encounter street level changes (SCV=0.5501), or a minor speed reduction (SCV=0.9091) when the movement takes place over rough surfaces (see Álvarez et al. (2018) for details).

Because the spatial distribution of the population depends on the time of the day, we consider two scenarios: nighttime and daytime. The nighttime case is modeled using data of the demographic census disaggregated at the block level (INE, 2012).





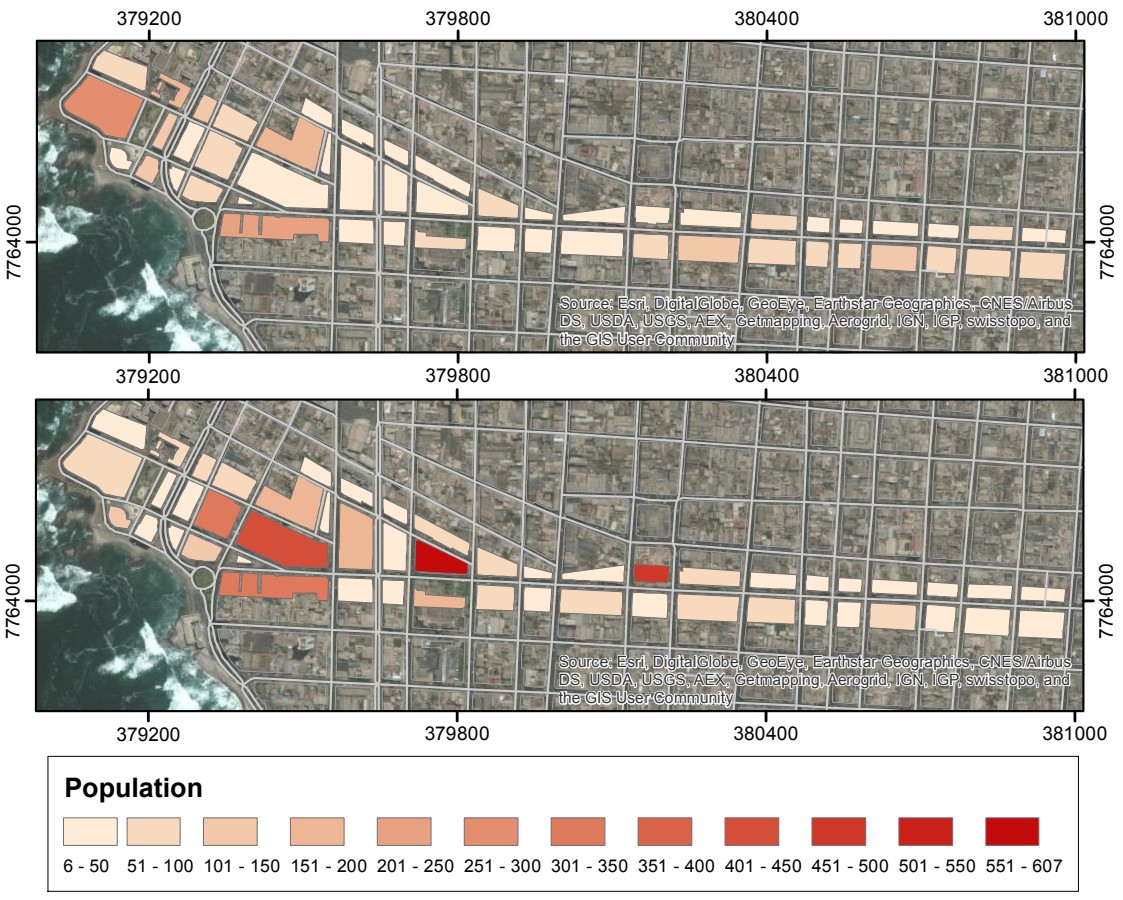

**Figure 5.** Population distribution in the studied area. Upper panel: nighttime scenario. Lower panel: daytime scenario.

The population distribution in the daytime scenario is mainly based on the origin-destination survey for the city of Iquique (SECTRA, 2010), but is complemented with information from the census, the permitted load of non-housing infrastructures (MINVU, 2016), and information related to the occupancy of educational establishments from the Ministry of Education. In
Figure 5 the population distribution is presented around the studied street for nighttime and daytime scenarios. Based on the data, the total number of agents is set to 3,171 in the nighttime scenario and 4,597 in the daytime scenario.

In order to take into account the variability in decision-making regarding the evacuation starting times of agents, we employ the Rayleigh cumulative probability distribution, which has been widely used in tsunami evacuation modeling (Mas et al., 2012; León and March, 2014, 2016; Solís and Gazmuri, 2017):

$$F(t) = 1 - \exp\left(\frac{-t^2}{2\sigma^2}\right) \tag{1}$$


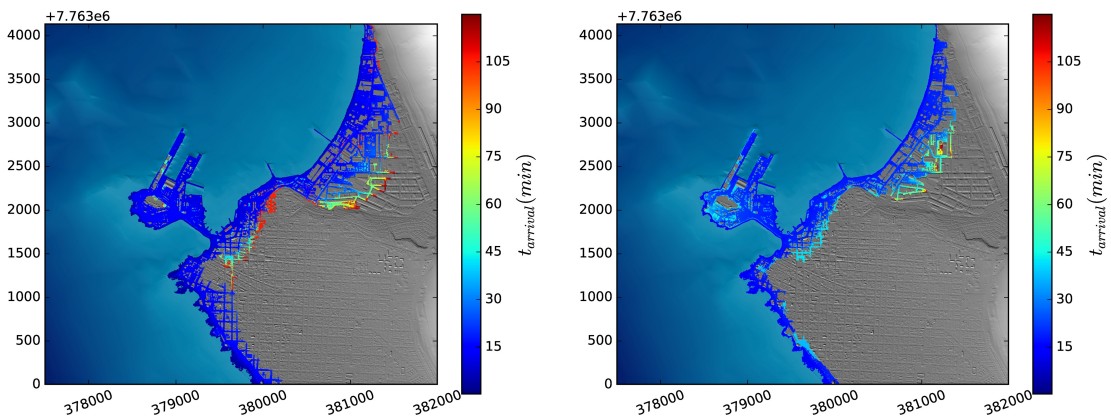

**Figure 6.** Tsunami arrival time maps for the modeled scenarios. Left: Mw 9.0 earthquake. Right: Mw 8.4 earthquake.

where $t$ is the time variable, and $\sigma$ is the scale parameter of the distribution; it is expressed in terms of the average starting time of evacuation $\bar{t}$ as $\sigma = \bar{t}\sqrt{2/\pi}$.

## 4 Results

### 4.1 Tsunami arrival times

In Figure 6 the tsunami arrival times for the two modeled scenarios are presented. The high topographic resolution (2 m) allows to represent the effect of the inundation through the streets of the city. It can be seen that in the Mw 9.0 scenario the first blocks of the city are flooded in less than 15 minutes, while in the Mw 8.4 scenario this area is flooded in 20 minutes. The most exposed areas are the port of Iquique, and the Duty Free Zone (ZOFRI), home of the commercial hub of the city, located north of the port. The evacuation analysis herein is focused around the Zegers Street, where tsunami inundation appears

to take between 15 and 25 minutes to penetrate six blocks. This street is also one of the routes of greatest concern due to the presence of urban micro-vulnerabilities (Álvarez et al., 2018). Therefore, different evacuation scenarios are modeled in detail, considering all the streets that feed the Zegers evacuation route as the modeling zone (see Figure 1).

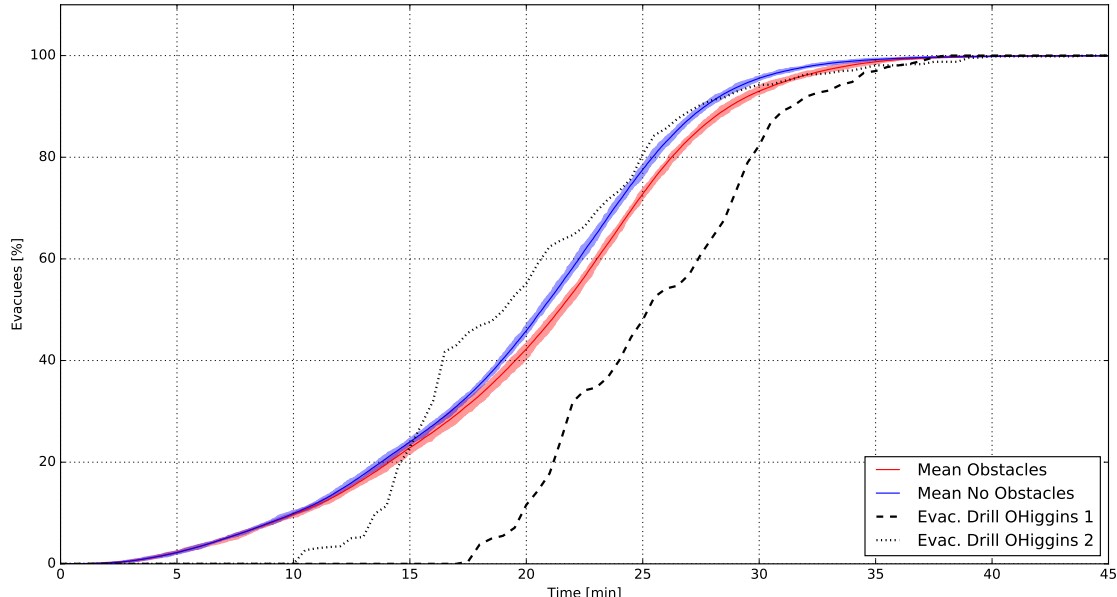

**Figure 7.** Validation of evacuation curves for the daytime scenario using a mean departure time of 3 min compared with two measurements from the August 8th 2013 drill. The ABM model is run with (red) and without (blue) urban micro-scale vulnerabilities - maximum and minimum limits of the simulation settings are shown and the center line corresponds to the average curve.

## 4.2 Validation of the Rayleigh evacuation curves

On the 8th of August 2013, the National Emergency Office (ONEMI) carried out a tsunami drill in the north of Chile as
part of the national preparedness plans. In the Tarapacá Region (Iquique is the regional capital), 76,000 people participated in the drill (Walker, 2013). During the drill, a research team of the Research Center of Integrated Disaster Risk Management (CIGIDEN) recorded the progress of the drill through a count of arrivals at meeting points on designated evacuation routes every 30 seconds. The counting of evacuees was used to build evacuation curves (Solís and Gazmuri, 2017). Here we use the evacuation curves built for the O'Higgins street since data on the evacuation of the Zegers street could not be properly
completed. Two independent measurements for evacuees are available for this street.

It is observed that the final evacuation times are close to 35 minutes, with no major differences between the simulations and the drill measurements. However, the shapes of the evacuation curves differ, especially for the initial times, which could be explained by a low turnout of people located in areas closer to the safe zone, whose times of arrivals are shorter. Nonetheless, the evacuation curves obtained by the ABM are considered a fair representation of the process.

## 4.3 Evacuation process under the modeled tsunami scenarios

We present the results of the ABM evacuation simulations for the Zegers street considering i) nighttime and daytime scenarios, ii) with and without the effect of urban micro-scale vulnerabilities, and iii) considering three times for the initiation of the





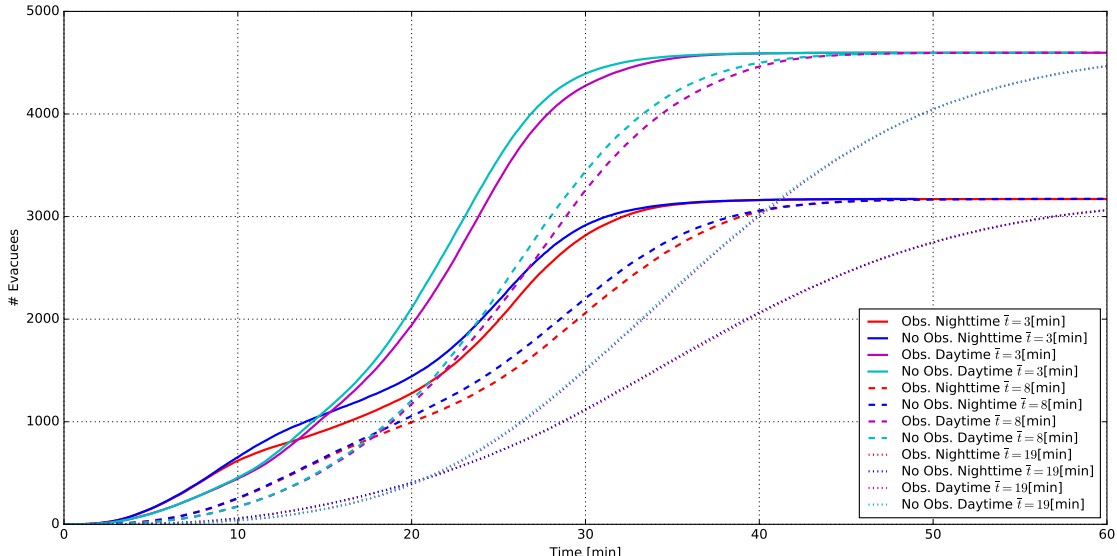

**Figure 8.** Average evacuation curves for the 12 modeled scenarios considering the presence and absence of urban micro-scale vulnerabilities, different departure times, and nighttime and daytime spatial distribution for agents.

agents' evacuation. We assume a mean departure time of $\bar{t} = 3$ minutes for an optimistic scenario where the agents start the evacuation immediately after the seismic shake; a mean departure time of $\bar{t} = 8$ minutes assuming that most of the agents start

evacuating once the tsunami alarm is activated; and a pessimistic scenario where the agents start evacuating at $\bar{t} = 19$ minutes when the first tsunami waves are visible.

Thus, a total of 12 combinations are simulated to assess the evacuation process. For each scenario, 30 ABM simulations are run to capture the variability in the initial spatial distribution of the agents within the blocks adjacent to the Zegers street, the speed functions and departure times; by doing so we expect to achieve an error of less than 1% in the number of evacuees at

each time, using a confidence interval of 95% (Byrne, 2013; Ritter et al., 2011).

In Figure 8 we present the average evacuation curves for each modeled scenario, in which the upper curves correspond to the results of the daytime scenarios (with more agents to be evacuated) and the lower curves to the nighttime scenarios. The total time needed to completely evacuate the agents to safe zones does not undergo significant variations in response to changes in the population distribution, since the differences in the distances that the agents have to travel along the evacuation route

and the agent speed functions show a moderate variability within the simulations. On the other hand, the differences between scenarios where agents encounter urban micro-scale vulnerabilities are also mild, but may not be negligible. The maximum instantaneous difference in the number of evacuees (with and without urban micro-scale vulnerabilities) for the case of $\bar{t} = 3$ minutes is nearly 250 agents in the daytime scenario, and 180 agents in the nighttime scenario (see Figure 10). When the mean departure time increases, the maximum instantaneous difference in evacuees tend to decrease since the route appears to be less





**Table 1.** Percentage of agents reached by the tsunami under the Mw 8.4 scenario assuming mean tide and high tide (in parenthesis).

|  | Percentage | | |
| --- | --- | --- | --- |
|  | $\bar{t} = 3$ min | $\bar{t} = 8$ min | $\bar{t} = 19$ min |
| Nighttime no urban micro-scale vulnerabilities | 0.0 (0) | 1.5 (3.5) | 32.6 (47.8) |
| Nighttime with urban micro-scale vulnerabilities | 0.0 (0) | 1.7 (3.6) | 32.5 (47.3) |
| Daytime no urban micro-scale vulnerabilities | 0.0 (0) | 1.5 (3.9) | 33.0 (57.7) |
| Daytime with urban micro-scale vulnerabilities | 0.0 (0) | 1.6 (4.0) | 33.0 (56.6) |

**Table 2.** Percentage of agents reached by the tsunami under the Mw 9.0 scenario assuming mean tide and high tide (in parenthesis).

|  | Percentage | | |
| --- | --- | --- | --- |
|  | $\bar{t} = 3$ min | $\bar{t} = 8$ min | $\bar{t} = 19$ min |
| Nighttime no urban micro-scale vulnerabilities | 0.0 (0.2) | 14.4 (22.3) | 62.3 (72.1) |
| Nighttime with urban micro-scale vulnerabilities | 0.1 (0.2) | 14.5 (22.3) | 62.1 (72.2) |
| Daytime no urban micro-scale vulnerabilities | 0.0 (0.1) | 9.8 (16.2) | 58.4 (68.7) |
| Daytime with urban micro-scale vulnerabilities | 0.0 (0.1) | 9.7 (16.0) | 58.0 (68.6) |

congested; as expected, urban micro-scale vulnerabilities have more influence on the evacuation process as the route becomes more crowded.

The effects of the modification of the mean departure time on the evacuation curves can also be observed in Figure 8. When the agents start evacuating with a mean departure time of $\bar{t} = 3$ minutes, most of the evacuees reach the safe zone within 30 minutes. On the other hand, when the agents' starting evacuation time is set to $\bar{t} = 8$ minutes, the time needed to get to the 225 safe zone is around 50 minutes. In the pessimistic scenario when $\bar{t} = 19$ minutes, more than an hour is needed to complete the evacuation process.

We now perform the analysis of the evacuation process considering the tsunami inundation scenarios. Since the ABM simulation gives the position of each agent over time with a 1-second resolution, and the tsunami inundation is modeled also at high spatial and temporal resolution, it is possible to estimate the number of agents that are reached by the tsunami at each 230 time. We show a snapshot of the ABM and tsunami simulations in Figure 9. The two tsunami scenarios (Mw 9.0 and Mw 8.4) are run for a mean tide level and a high tide level. In Table 1 and 2 we summarize the percentage of agents (over the total) that are reached by the tsunami under the different studied cases. For the two tsunamigenic scenarios, when the agents' mean departure time is set to $\bar{t} = 3$ minutes, the percentage of agents reached by the tsunami is zero or very low (less than 0.2 %), thus demonstrating that a prompt evacuation starting immediately after the earthquake is paramount to save lives.

When the mean departure time is $\bar{t} = 8$ minutes, the percentage of agents caught by the tsunami increases significantly, with values between 1.5 % and 4.0 % for the Mw 8.4 earthquake, and between 16.0 % and 22.3 % for the Mw 9.0 scenario. It is important to note that in the nighttime case more agents are reached by the tsunami, specially in the Mw 9.0 earthquake. The latter is due to the residential nature of the housings that are located very close to the coastline which implies that a higher


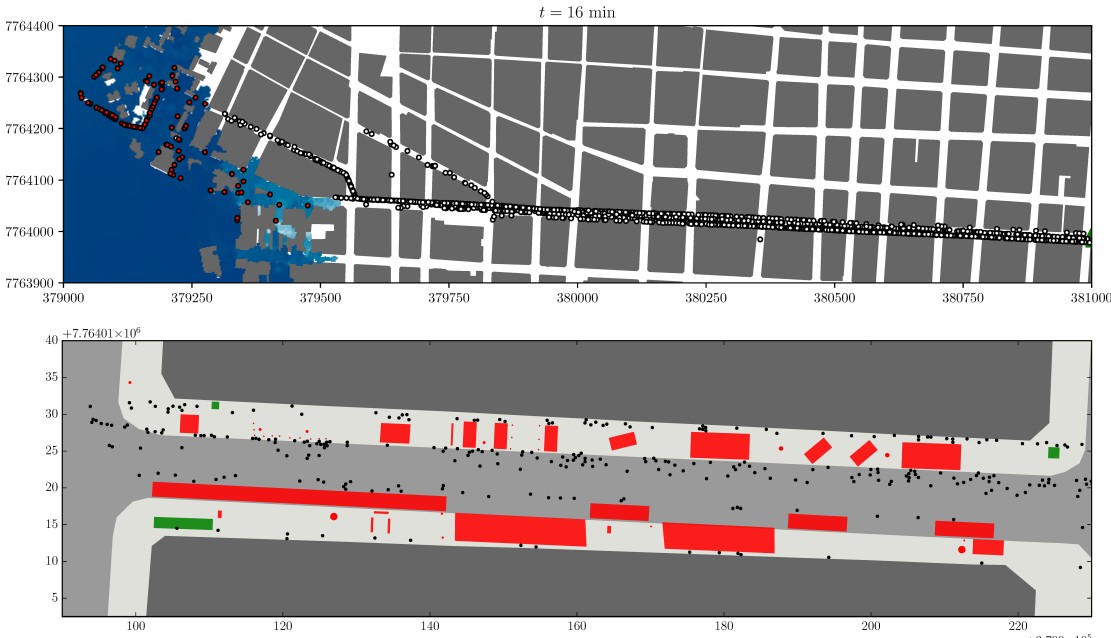

**Figure 9.** Snapshots of the ABM simulation for the Zegers evacuation route. Upper panel: The first tsunami arrivals are shown for the Mw 9.0 scenario at time $t = 16$ minutes; agents colored in white are in the process of evacuating while those colored in red have been reached by the tsunami. Lower panel: Example of a street segment and the representation of urban micro-vulnerabilities: obstacles that impose a speed reduction are depicted in green, while the ones that block the agents' passage are colored in red.

number of agents are exposed to the tsunami within the first blocks during nighttime (see Figure 5). No significant differences
between the cases with and without urban micro-scale vulnerabilities are observed, while the percentage of agents caught by
the tsunami is much higher for the simulations under high tide.

In the pessimistic case with a mean evacuation departure time of $\bar{t} = 19$ minutes, the percentage of agents reached by the
tsunami shows a radical increase. For the Mw 8.4 scenario, nearly 30 % of the agents are caught by the tsunami for the mean
tide case, while around 50 % are reached when the tsunami simulation is performed under high tide conditions. For the Mw 9.0
earthquake, the percentage of agents that are reached by the inundation is around 60 % and 70 % under the mean and high tide
case, respectively. It is worth noting for the Mw 9.0 more agents are caught by the tsunami under the nighttime scenario (though
only 3-4 % more), thus consistent with the previous results, but the situation is the opposite for the Mw 8.4 earthquake under
high tide, where more agents are reached during daytime. The latter maybe due to the arrival of the first waves within 20-25
minutes over the blocks having a high rate of occupancy in daytime (see Figure 5). On the other hand, again, the influence of
the urban micro-scale vulnerabilities is not decisive for the evacuation process.





## 5   Discussion

The analysis we conducted did not show overall significant influences of the urban micro-scale vulnerabilities on the evacuation processes. However, it was shown that the latter would depend on the level of congestion, so we attempt now to measure their influence by computing the differences in the number of evacuees at each time for simulations with and without their presence.

To quantitatively assess this, we perform ABM simulations with $\bar{t} = 3$ minutes and increasing the total number of agents in the daytime and nighttime scenarios by factors of 1.5, 2, 2.5, and 3. In Figure 10 we show the differences in the number of evacuees for the ABM simulations with and without urban micro-scale vulnerabilities. In the daytime scenario the maximum difference in evacuees increases from 250 to 550 (2.2 times higher) when the number of agents is doubled, and to 720 (+2.9 times) when it is tripled. For the nighttime scenario the maximum difference of 180 evacuees increases to 290 (+1.6 times) and 370

(+2.06 times), when the total number of agents is doubled and tripled, respectively. It is thus shown that as the evacuation route becomes more crowded, the delays that the presence of urban micro-scale vulnerabilities induce in the evacuation processes increase.

The relatively small influence of the urban micro-scale vulnerabilities on the outcomes of an evacuation process in the study area could rapidly change if other real-world conditions (consistent with observations during tsunami emergencies in Chile) are included in the model. For instance, while our analysis considered that agents could use both the sidewalks and the road as a movement area, tsunami evacuations in other cities have shown that this was unfeasible due to full road occupancy provoked by automobile traffic congestion (León et al., 2022). In such a case, where evacuees might be confined to the sidewalks, urban micro-scale vulnerabilities' impact on their speeds could be much higher.

The expected impact of the urban micro-scale vulnerabilities on pedestrian evacuation time could also change depending on the chosen modelling technique and how it assesses the effect of the agents' agglomeration density. For instance, queuing models focused on evacuation (e.g. Lämmel et al., 2010) typically simulate the urban realm as a network of links and nodes throughout which agents move. Each link has a definite capacity that limits the number of evacuees that can simultaneously be within it, and, therefore, incoming agents must wait to enter the link if there is not enough available space. Other network-based models (Goto et al., 2012; Wang and Jia, 2021) use the spatial properties of each street link (width, length) to estimate its

area and then apply a continuous function to assess the impact of the agents' density on their pedestrian speed within it. León et al. (2021) proposed a raster-based agent-based model where each agent assesses the density of agents located in its next destination cell along its escape path. If this density surpasses 1.3 agent/m$^2$, the agent changes direction to avoid it (therefore its evacuation path and time extends). In future work, different agent-based models could be used to assess the same evacuation scenario and micro-scale parameters in Iquique, and subsequently perform a sensitivity analysis to examine how the results

change among them.

The urban micro-scale vulnerabilities approach could also be used to assess how post-earthquake urban scenarios could dynamically affect evacuation. In this respect, existing models (Aguirre et al., 2018) use the material characteristics of the building stock to estimate the amount of seismic-related damage that a given tsunamigenic earthquake can provoke in a city. In turn, this analysis could lead to an assessment of the likely amount of debris to be deposited along the streets. In the agent-





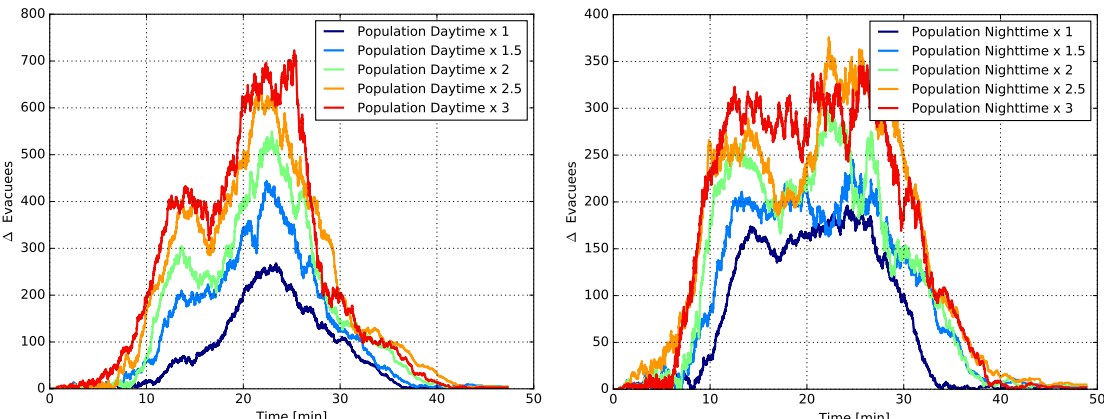

**Figure 10.** Quantification of the impact of urban micro-vulnerabilities in the evacuation process. Difference in the number of evacuees when the total number of agents is increased by factors of 1.5, 2, 2.5, and 3. Left panel: daytime scenario. Right panel: nighttime scenario.

based model, this debris could be included as new micro-scale vulnerabilities that further delay the evacuation (e.g. Castro et al., 2019).

## 6 Conclusions

Recent progress in tsunami sciences and disaster risk management have provided better Early Warning Systems, protocols, and plans to improve the evacuation response of the population, and ultimately save lives. However, the analysis of the impact that

the so-called 'urban micro-scale vulnerabilities' (which result from a poor maintenance or practices along evacuation routes) may have on evacuation processes have received only little attention in the specialized literature. We have thus conducted here a series of Agent-Based Model (ABM) simulations to assess the effects that the urban micro-scale vulnerabilities identified in the city of Iquique in Chile, could produce in the number of evacuees that may escape from the tsunami along one of the main evacuation routes in the city. Two demanding tsunamigenic scenarios with arrival times between 15-20 minutes were

considered to assess the evacuation process along the Zegers street located downtown of the city. Daytime and nighttime scenarios were considered and three different average evacuation starting times were assumed in the evacuation simulations.

The most important variable controlling the number of agents caught by the tsunami was shown to be the average departure time for the evacuation. A self-evacuation immediately after the earthquake (3 minutes) would allow to save almost all agents in the studied area under all the considered scenarios. On the contrary, a delayed evacuation with an average time of 19 minutes,

would result in a large increase (between 30% and 70%) in the number of agents reached by the tsunami.

The differences in evacuation times with and without the presence of urban micro-scale vulnerabilities in the Zegers street were mild but not negligible, and are more important as the average departure evacuation time is reduced (and the evacuation route is more crowded). Nevertheless, their effect in the number of agents caught by the tsunami was not decisive in the studied

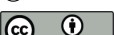



case. Through additional simulations that considered doubling and tripling the total number of agents, it was shown that as the

evacuation route becomes more crowded, the delays that the presence of urban micro-vulnerabilities induce in the evacuation processes may increase significantly. Thus, the relatively small effect of the micro-scale vulnerabilities on the outcomes of an evacuation process in the study area could rapidly change if other real-world conditions, such as a full road occupancy provoked by automobile traffic congestion, are considered. When evacuees might be confined to the sidewalks, urban micro-scale vulnerabilities' impact on their speeds could be much higher.

*Code availability.* The codes used in this work can be made available upon request to the authors.

*Data availability.* The data used in this work can be made available upon request to the authors, but we must comply with third-party restrictions (i.e. topo-bathymetric data).

*Author contributions.* R. Cienfuegos and J. León conceived, conceptualized, and supervised the research work. R. Cienfuegos organized and edited the manuscript. G. Álvarez and S. Castro compiled the required data, implemented the ABM evacuation model, and produced the

outputs and graphics. A. Urrutia and G. Álvarez prepared the required data, implemented the tsunami model, and run the tsunami simulations, and prepared the outputs and graphics. All co-authors participated in the analysis of the results and discussions, and contributed to the edition of the final manuscript.

*Competing interests.* The authors declare that no competing interests are present.

*Acknowledgements.* This research work was undertaken thanks to the support of the Centro de Investigación para la Gestión Integrada

del Riesgo de Desastres (CIGIDEN), under the grant number ANID/Fondap/1522A0005. We acknowledge the support of the National Hydrographic and Oceanographic Service of the Chilean Navy (SHOA), and the JICA-JST Satreps program for providing the high resolution topo-bathymetric data requiered for the tsunami simulations. We also thanks the National Service for Disaster Prevention and Disaster Preparedness (SENAPRED) for the support they provided during the tsunami evacuation drills undertaken in the Iquique city.



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
