# Peer review of "Analysis of the effects of urban micro-scale vulnerabilities on tsunami evacuation using an Agent-Based model. Case study in the city of Iquique, Chile"

_Natural Hazards and Earth System Sciences, 2023_

## Author Comment (AC1)

**Referee 1**

We acknowledge the careful review of referee 1 and the suggestions he/she provides for improving our manuscript. Below we point out the answers to all his queries and the changes introduced in the amended document.

Until recently tsunami numerical studies were primary concentrated on the hazard side less focusing on damage and loss assessment. Rapid development of computational facilities and corresponding numerical codes coupled to collection of high precision and high resolution topo-bathymetric and exposure data allow nowadays detailed and reliable simulations of tsunami inundation and impact scenarios. Tsunami modeling community is putting more and more efforts in exploring methodologies to assess potential tsunami damages and losses and, consequently, factors controlling them in order to propose effective counter-measures. In this respect, current manuscript is an important step forward. Authors present and evaluate quantitative coupled tsunami simulation / agend-based methodology to assess the effectiveness of evacuation in presence of artificial obstacles along evacuation routes. The Manuscript is concise, well structured and easy to read. I do not have any major objections and recommend publishing after minor revisions.

First of all I have two comments/suggestions regarding the methodology. They both concern daytime to nighttime variations.

(1) Authors analyze the effect of „micro-scale vulnerabilities", i.e. obstacles along evacuation routes. These may include e.g. restaurant infrastructure – tables, chairs, – or shop's advertisement objects placed directly on street sidewalks. This obstacles are usually removed from the streets during the night time, means the distribution of obstacles may be also time-dependent.

The referee's comment is correct. There might be some nighttime/daytime dependance over some of the obstacles. In the present analysis we have not incorporated this dimension, but it could be an interesting research avenue for future work. We have included comments on these lines in the discussion section.

(2) Also the average evacuation starting time should probably be different for day and night. Just because people at night, while being awakened by strong shaking or sirens, won't be so effective in starting their evacuation as it could be during the day.

It could definitively be the case, we have not included in the simulation specific modifications of the evacuation departure time taking into consideration nighttime/daytime conditions, but we studied the model results using three different departure times covering a wide range of possibilities. We have included a comment regarding this matter in the discussion section.

I do not ask to implement these options in the present study but just to discuss their feasibility for future studies.

Other minor remarks.

- References to Satake 2014a and b are the same.

Since we have re-oriented the introduction and the literature review following the suggestions of referee 2, this reference is no longer included in the article.

- L102: It is better to define „urban friction rate" here, in a following sentence.

Thanks for this suggestion which helps to clarify this concept. We have added a sentence there and a specific quote.

- L159: Maybe refer to Fig. 9 which illustrates distribution of MSV along the evacuation route.

We have referred to the Figure (now Fig. 10) for clarity following the suggestion of the referee.

- Sect.3.4: which tool was used for the agend-based modeling? Authors describe components of the ABM in the first paragraph of this section, but do not mention the modeling tool.

The model used in this work was developed and validated in the article "Validation of an agent-based building evacuation model with a school drill" by Poulos et al. (2018). This is now clarified in the amended version of the manuscript.

- Fig.6: This Figure could be much more informative. First of all, the current figure size does not allow to recognize enough details – I suggest to make both plots much larger: at least half page each. One should be able to see the details of the inundation patterns. For that please also „resolve" the blue color: now it encompasses arrival times from 0 to 45 minutes – the most important time span unfortunately not resolved at all. High spatial and temporal resolution would also help to understand the abrupt arrival time gradient (from less than 15 to more than 90 minutes) observed at the left plot. Additionally, I think, it is necessary to show the distribution of the initial tsunami wave height offshore. That would help to interpret extreme short arrival times. This can be done using larger-scale inlets and/or integrated into the image. Currently the blue ocean is confusing: is it bathymetry? Or arrival times as well? In the latter case they seem to be inconsistent with dark-blue (=earlier) arrivals on land.

Thank you for this suggestion. We have re-worked the former Figure 6 to give more details and better visualization of the tsunami inundation and arrival times (now Fig. 7). We have also added a new Figure (Figure 5) to represent the initial conditions for the tsunami scenarios and the resulting time series at the tidal gauge located near the port of Iquique.

Fig.7: Empirical evacuation curves start from 10 or 17 minutes. Were there absolutely no people in the vicinity of the safe zones? With 76000 drill participants it may seem indeed strange. Just looking at this Figure, I would propose to critically re-assess the methodology of the average starting time to make it more compatible to observations.

Thank you for your comment. Some factors could contribute to explaining the differences between the evacuation rates as measured during the 2013 evacuation drill and those shown by our model. The first one is the low recording of evacuees by CIGIDEN during the drill, in comparison to the overall number of participants: according to Solís and Guzmán (2017), while roughly 76,000 people participated in the Iquique drill, only 12,658 (16.7%) were registered in the examined assembly points (eleven). Moreover, as participation in the exercise was not mandatory, it is likely that the recorded participants' departure locations were unevenly distributed across the city, with a more significant participation rate in areas close to the sea (therefore with longer evacuation times), and by specific institutions (primary and secondary schools) that commonly take part in drills and that in Iquique tend to be located in these coastal areas, unlike people that live, work or study close to the assembly points in higher grounds.

The second factor that could have contributed to delayed arrivals at the assembly areas might be related to the fact that in Chile evacuation drills require the population to wait for 2-3 minutes to begin the evacuation after the warning is released, to resemble the time length of a large, tsunamigenic earthquake, that would not allow people to move while it is still occurring. Unlike this delay, the Rayleigh distribution allows a few agents to begin to move as soon as the modelled earthquake begins.

While we acknowledge that the former Figure 7 (Fig. 8 in the amended manuscript) shows time differences across the beginning times of the arrival of evacuees at the assembly areas, we also point out that the modelled times are capable of reflecting with significant accuracy the required times for total evacuation, as measured during the drill (roughly 35 minutes), and to show similar evacuation rates to those collected during the exercise, for the evacuation period after 17 minutes of evacuation. Having said that, it is also important to underline that in this article we do not attempt to deliver accurate modelling of the 2013 evacuation but rather to examine the potential impact of micro-vulnerabilities on evacuation times, within the context of our evacuation model.

We have modified the title of this section from "Validation of the Rayleigh evacuation curves" to "Assessment of the modeled evacuation curves". We have also incorporated the points we described here.

Fig. 8 and L225: Looking at the Figure 8, I cannot agree that for the case of „8-min starting time", time needed to get to the safe zone is 50 minutes. Actually comparing daytime dashed and solid lines, I would evaluate the time difference between them as „less than 10 minutes". Thus, if at t=30 min solid lines reach the level of ca. 4400 evacuees, the same level will be reached by dashed lines at t~38 min. Therefore, time difference between „3 and 8-min" scenarios seems to be much less than imposed 20 (50 vs 30) minutes.

Thank you for your comment. We acknowledge here that there is a problem with the writing of our paragraph, as it is comparing two different thresholds. In the case of "continuous" lines (i.e., average departure time = 3 min), the manuscript (1st submission) says "*most of the evacuees reach the safe zone within 30 minutes*" (line 224), therefore establishing a reference threshold (30 minutes) that does not correspond to the required time to evacuate the totality of its evacuees (roughly 40 minutes). On the other hand, for the "dashed" lines (average departure time = 8 min), the paragraph points out that "*the time needed to get to the safe zone is around 50 minutes*" (line 225). This sentence refers to the *totality* of the evacuees; therefore, if we compare both required times for "total" evacuation, the time difference between them is around 10 minutes, as the reviewer correctly pointed out. In the new submission, we modified the paragraph accordingly.

---

## Author Comment (AC2)

**Referee 2**

We acknowledge the careful review of referee 2 and the suggestions he/she provides for improving our manuscript. Below we point out the answers to all his queries and the changes introduced in the amended document.

A review on the manuscript "Analysis of the effects of urban micro-scale vulnerabilities on tsunami evacuation using and Agent-Based model. Case study in the city of Iquique, Chile" written by Rodrigo Cienfuegos, Gonzalo Álvarez, Jorge León, Alejandro Urrutia, and Sebastián Castro

The manuscript presents an analysis of the challenges surrounding tsunami evacuations, with a focus on Iquique, Chile. Using an Agent-Based Modeling (ABM) and tsunami inundation simulations, the study quantifies the evacuation processes resulting from urban micro-vulnerabilities and other factors. This research has a significant contribution, emphasizing the necessity of addressing such vulnerabilities in disaster management. The manuscript can be accepted for publication after a minor revision.

Here are some of the suggestions for improvement. My main concern is on the introduction part.

There is a typo in the title. "and"- >"an"

We have corrected this typo, thank you.

The introduction lacks a clear and explicit statement of the article's main thesis or research question. It would greatly benefit from a more precise outline of the study's objectives, the specific questions it aims to address, and a clear description of how it contributes to existing research in comparison to previous studies.

While historical context is important, the introduction dedicates a substantial portion of its content to discussing past tsunamis and their consequences. This historical background information could be streamlined to allow for a more focused and concise introduction that directly introduces the study's subject matter.

We have re-written the introduction following the lines suggested by the referee. We hope it is now much focused and concise.

In Section 4.2, titled "Validation of the Rayleigh evacuation curve," the term "validation" might not be entirely accurate. Rather, it appears to be a comparison to previous evacuation drills. The discrepancy in the average starting times between the ABM simulations and the actual drills (3 minutes vs. 10 to 17 minutes) raises questions about the model's validation. Additionally, the duration required to achieve close to 100% evacuation seems to be faster in the case of the drills, suggesting that the ABM model may

not be accurately validated for this aspect. A more precise description of the model's performance and limitations in comparison to real-world data would be beneficial.

Thank you for this comment. Referee 1 also raised questions regarding this issue. We have changed the title of the section to better reflect that it is not a validation but an assessment of the model performance. We have also attempted to explain in the amended manuscript the observed differences between the model and the drill. There are at least two factors that we could identify:

The first one is the low recording of evacuees by CIGIDEN during the drill, in comparison to the overall number of participants: according to Solís and Guzmán (2017), while roughly 76,000 people participated in the Iquique drill, only 12,658 (16.7%) were registered in the examined assembly points (eleven). Moreover, as participation in the exercise was not mandatory, it is likely that the recorded participants' departure locations were unevenly distributed across the city, with a more significant participation rate in areas close to the sea (therefore with longer evacuation times), and by specific institutions (primary and secondary schools) that commonly take part in drills and that in Iquique tend to be located in these coastal areas, unlike people that live, work or study close to the assembly points in higher grounds.

The second factor that could have contributed to delayed arrivals at the assembly areas might be related to the fact that in Chile evacuation drills require the population to wait for 2-3 minutes to begin the evacuation after the warning is released, to resemble the time length of a large, tsunamigenic earthquake, that would not allow people to move while it is still occurring. Unlike this delay, the Rayleigh distribution allows a few agents to begin to move as soon as the modelled earthquake begins.

While we acknowledge that the former Figure 7 (Fig. 8 in the amended manuscript) shows time differences across the beginning times of the arrival of evacuees at the assembly areas, we also point out that the modelled times are capable of reflecting with significant accuracy the required times for total evacuation, as measured during the drill (roughly 35 minutes), and to show similar evacuation rates to those collected during the exercise, for the evacuation period after 17 minutes of evacuation. Having said that, it is also important to underline that in this article we do not attempt to deliver accurate modelling of the 2013 evacuation but rather to examine the potential impact of micro-vulnerabilities on evacuation times, within the context of our evacuation model.

In Section 5, L257 states "In Figure 10 we show the differences in the number of evacuees for the ABM simulations with and without urban micro-scale vulnerabilities." However, Figure 10 does not show the results "with and without urban micro-scale vulnerabilities".

Thank you for pointing out this. The title of the Figure was not clear. Indeed, the vertical axis show the temporal difference between the number of evacuee when the simulation is

run without and with urban micro-vulnerabilities. We have clarified this in the title and in the text.

L17 "Tsunami research has come a long way after this event" -> unclear what this implies

Since the introduction was re-organized, this phrase has been deleted.

L54 "possible only" -> "only possible"

The introduction has been re-written.

L71 "The largest reported earthquakes in recent centuries in the area" -> "The most notable seismic events recorded in the region in recent centuries include"

This paragraph has been re-worked.

L236 "16 %" -> Is it 9.7 % ?

You are right, we have corrected this value in the amended version of the manuscript.

Figure 3&4 Recommend to plot the location of DARTs in Figure 3.

We have updated the Figure 3 to include the locations of the Dart buoys.

Figure 6. Change the color scheme. It is difficult to distinguish the arrival time between 15 min and 20 min.

Also following the comments of referee 1, we have improved the former figure, 6 and added a new one (Figure 5 in the amended manuscript) to show the initial sea surface deformation scenarios and wave height time series).